# Molecular Dynamics of Cobalt Protoporphyrin Antagonism of the Cancer Suppressor REV-ERBβ

**DOI:** 10.3390/molecules26113251

**Published:** 2021-05-28

**Authors:** Taufik Muhammad Fakih, Fransiska Kurniawan, Muhammad Yusuf, Mudasir Mudasir, Daryono Hadi Tjahjono

**Affiliations:** 1School of Pharmacy, Bandung Institute of Technology, Jalan Ganesha 10, Bandung 40135, Indonesia; taufikmuhammadf@unisba.ac.id (T.M.F.); siska@fa.itb.ac.id (F.K.); 2Department of Pharmacy, Faculty of Mathematics and Natural Sciences, Universitas Islam Bandung, Jalan Rangga Gading 8, Bandung 40116, Indonesia; 3Department of Chemistry, Faculty of Mathematics and Natural Sciences, Universitas Padjadjaran, Jalan Raya Bandung Sumedang Km 21, Sumedang 45363, Indonesia; m.yusuf@unpad.ac.id; 4Department of Chemistry, Faculty of Mathematics and Natural Sciences, Universitas Gadjah Mada, Sekip Utara BLS 21, Yogyakarta 55281, Indonesia; mudasir@ugm.ac.id

**Keywords:** breast cancer, REV-ERBβ, porphyrin, nuclear receptor corepressor (NCoR), molecular dynamics (MD) simulation, porphyrin

## Abstract

Nuclear receptor REV-ERBβ is an overexpressed oncoprotein that has been used as a target for cancer treatment. The metal-complex nature of its ligand, iron protoporphyrin IX (Heme), enables the REV-ERBβ to be used for multiple therapeutic modalities as a photonuclease, a photosensitizer, or a fluorescence imaging agent. The replacement of iron with cobalt as the metal center of protoporphyrin IX changes the ligand from an agonist to an antagonist of REV-ERBβ. The mechanism behind that phenomenon is still unclear, despite the availability of crystal structures of REV-ERBβ in complex with Heme and cobalt protoporphyrin IX (CoPP). This study used molecular dynamic simulations to compare the effects of REV-ERBβ binding to Heme and CoPP, respectively. The initial poses of Heme and CoPP in complex with agonist and antagonist forms of REV-ERBβ were predicted using molecular docking. The binding energies of each ligand were calculated using the MM/PBSA method. The computed binding affinity of Heme to REV-ERBβ was stronger than that of CoPP, in agreement with experimental results. CoPP altered the conformation of the ligand-binding site of REV-ERBβ, disrupting the binding site for nuclear receptor corepressor, which is required for REV-ERBβ to regulate the transcription of downstream target genes. Those results suggest that a subtle change in the metal center of porphyrin can change the behavior of porphyrin in cancer cell signaling. Therefore, modification of porphyrin-based agents for cancer therapy should be conducted carefully to avoid triggering unfavorable effects.

## 1. Introduction

Porphyrin-based agents can be an alternative choice for cancer therapy because they have selective cytotoxicity against tumor cells [1]. Porphyrins are a family of organic ring molecules including Heme (Iron protoporphyrin IX), the pigment in red blood cells, an essential molecule for living aerobic organisms, plays a major role in gas exchange, mitochondrial energy production, antioxidant defense, signal transduction, and they represent one of the oldest and most widely studied chemical structures in nature and in biomedical applications [2,3,4]. Moreover, Heme is also contained in the superfamily of enzymes Cytochromes P450 (CYPs) which plays a role in supporting oxidative, peroxidative, and reductive metabolism of endogenous and xenobiotic substrates [5,6]. P450s are endoplasmic reticulum (ER)-anchored hemoproteins which responsible for the metabolism of numerous endogenous and foreign compounds. As the prosthetic moiety of all P450s, Heme is responsible for the remarkable and often exquisite, catalytic prowess of these enzymes [7].

The optimal dose of a porphyrin-based agent is lethal to tumor cells while minimizing damage to adjacent normal tissue [8]. Therefore, multi-modal porphyrin-based agents have great potential for use in cancer imaging and therapy [9]. Porphyrins have been particularly useful in photodynamic therapy and fluorescence imaging of cancer because of their tumor avidity and favorable photophysical properties, such as long wavelength absorption and emission, easy derivatization, high singlet oxygen quantum yield, and low in vivo toxicity [10,11]. In addition, porphyrins are excellent metal chelators that form highly stable metallocomplexes, making them efficient delivery vehicles for radioisotopes [12]. Several studies have investigated the characteristics of various porphyrin-based probes and their clinical applications in cancer imaging and therapy [10]. Porphyrin-based agents can increase the signal-to-background ratio in tumor imaging, allowing better detection of tumor tissues and better preservation of healthy tissues during therapy [13].

The binding of the porphyrin Heme to the nuclear receptor REV-ERBβ leads to decreased proliferation in various cancer cells [14]. REV-ERBβ, along with REV-ERBα, is a member of the REV-ERB family of nuclear receptors. Although REV-ERBβ is structurally and functionally similar to REV-ERBα, it has a unique role in the regulation of circadian rhythms, which in turn affect a variety of diseases including cancer [15]. Moreover, REV-ERBβ is overexpressed in breast cancer cells, accounting for more than 95% of the total REV-ERB mRNA [15]. In addition, REV-ERBβ is dominantly expressed in several cancer cell lines, whereas REV-ERBα is the dominant form in corresponding normal tissues [16].

The binding of Heme to REV-ERBβ (Figure 1a) induces recruitment of nuclear receptor corepressors such as NCoR, which in turn regulate diverse pathways ranging from Heme biosynthesis to circadian rhythms [17]. NCoR forms stable complexes with transcription factors that are deregulated in various cancers [18]. Synthetic agonists have been developed that mimic the ability of Heme to induce NCoR recruitment and corepressor-peptide binding to REV-ERBβ [19]. However, previous research could not predict the ability of porphyrin-based REV-ERBβ antagonists to repress NCoR recruitment [20].

Besides Heme, there are other ligands that bind REV-ERBβ, including cobalt protoporphyrin IX (CoPP) and zinc protoporphyrin IX (ZnPP). The only difference between Heme and CoPP and ZnPP is in the coordinated metal ions. CoPP and ZnPP are functional antagonists of REV-ERBβ, whereas Heme is the natural agonist of REV-ERBβ. CoPP and ZnPP block the transcriptional repressive functions of REV-ERBβ by preventing its binding to protoporphyrin IX [21]. The iron metal center of Heme has a good affinity for Histidine (His) and Cysteine (Cys) residues on REV-ERBβ. Cobalt has a lower affinity than iron, but it can still interact with and regulate the properties of REV-ERBβ [22]. Therefore, Heme and CoPP can be used as reference structures for the development of selective porphyrin-based agents for cancer therapy and imaging.

Figure 1a shows that the addition of small molecules to the REV-ERBβ induces enlargement of the ligand-binding domain (purple box) which may affect the recruitment of NCoR (red color). However, the REV-ERBβ-Heme and REV-ERBβ-CoPP have similar crystal structure (Figure 1b) but show different biological activity. To develop new porphyrin-based agents for early cancer detection, a preliminary study is needed to compare the molecular dynamics of Heme and CoPP binding to REV-ERBβ. In particular, it is important to determine how iron or cobalt in the center of protoporphyrin IX affects the porphyrin ring conformation, which might determine whether protoporphyrin IX acts as an agonist or an antagonist of REV-ERBβ.

## 2. Results

### 2.1. Initial Pose of the Ligand–Protein Complex

Heme and CoPP were docked into two different of REV-ERBβ crystal structures: Antagonist-REV-ERBβ and Agonist-REV-ERBβ. Antagonist-REV-ERBβ (PDB ID: 4N73) is a crystalline structure of REV-ERBβ that forms a complex with CoPP, whereas Agonist-REV-ERBβ (PDB ID: 3CQV) forms a complex with Heme. Molecular docking was performed to investigate the binding modes and affinities of protoporphyrin IX within the binding pocket of REV-ERBβ. Both crystal structures of REV-ERBβ are shown in Figure 2. 

The overlay poses of the ligand structure during re-docking process and docking simulation are provided in Figure 3. Heme had the greatest negative binding energy in both crystal structures of REV-ERBβ, with the binding energy values of −65.91 kJ/mol (Antagonist-REV-ERBβ) and −71.10 kJ/mol (Agonist-REV-ERBβ; Table 1). Those results are in agreement with the results of previous isothermal titration calorimetry (ITC) studies that found that the binding of CoPP (K_D_ = 2.56 µM) was weaker than that of Heme (K_D_ = 353 nM) [22]. The results reflect the atomic radius of each metal bound to the center of protoporphyrin. Iron has an atomic radius of 0.126 nM, whereas cobalt has atomic radius of 0.200 nM, indicating that the radius of the atom at the metal center plays a role in determining the affinity of the porphyrin complex for REV-ERBβ. 

To investigate the interactions between both forms of protoporphyrin IX and the two forms of REV-ERBβ, the binding modes of CoPP and Heme within the REV-ERBβ binding pocket were observed further. In general, CoPP and Heme had the same interactions with the amino acid residues of REV-ERBβ (i.e., Val383, Gly480, and Leu483); however, Heme had additional interactions with Cys384 and His568 [21]. Visualization of the docking pose with REV-ERBβ revealed an interaction differences in the bonding of the central metal atom (Figure 4). Those differences correspond with the crystal structures in that they were different in terms of the interactions with Cys384 and His568, which were coordinated by the metal center of the protoporphyrin IX ring. That phenomenon is likely to be the main source of the stronger binding affinity of Heme compared with that of CoPP. The effects of interactions involving the central metal atom might determine the agonist and antagonist functions of REV-ERBβ. 

### 2.2. The Binding Free Energy of Agonist and Antagonist Ligands against REV-ERBβ Ligand-Binding Domain (LBD)

The molecular mechanics Poisson-Boltzmann surface area (MM/PBSA) method was used to predict the binding free energy of the four protein-ligand complexes more accurately. The binding free energies of the complexes were more accurate than those observed in the docking study (Table 2). Furthermore, the energies that mostly contributed were electrostatic and van der Waals interaction. Overall, all four complexes had good stability in the molecular dynamic simulations. 

The graph of the total energy of the system shows the beginning of changes in the protein conformation (Figure 5a). The total energy increased from 25 ns to 30 ns during formation of the Antagonist-REV-ERBβ complex, whereas it increased from 0 ns to 5 ns during formation of the Agonist-REV-ERBβ complex (Figure 5b). These results show that the antagonist ligand greatly influenced the conformational changes of the target protein.

### 2.3. Structural Analysis from Molecular Dynamics Simulation

Molecular dynamics (MD) simulation was performed with the four complexes to gain more structural, dynamical, and energetic information about the stability of Heme and CoPP in the ligand-binding domain (LBD) of REV-ERBβ. Snapshots of the MD trajectories of the four complexes were visualized to observe the time-dependent position of the ligand and also to analyze the effect of the interaction on the structure of REV-ERBβ. Visualization of the complexes during the MD simulations is shown in Figure 6. The poses of Heme and CoPP in the binding site were generally not different from one another. The central metal atom of protoporphyrin IX was stable and remained in the center of the ring until the end of the simulation. Moreover, the protoporphyrin IX remained planar throughout the simulation, indicating its function to deliver the metal atom to the protein target. That is in agreement with the absorption spectral bands of Heme and CoPP, because a nonplanar conformation resulting from repulsive interactions among the peripheral substituents would induce dramatic redshifts of the electronic absorption spectral bands [23].

### 2.4. Dynamic Behavior of REV-ERBβ in the Antagonist Form

The root-mean-square deviation (RMSD) represents the stability of the molecular structure throughout the simulation. MD simulations of the Heme and CoPP ligands in complex with Antagonist-REV-ERBβ were conducted to observe the behavior of both ligands. Figure 7a shows the RMSD of Antagonist-REV-ERBβ in the presence of Heme and CoPP. The average RMSD values for Heme and CoPP were 3.49 Å and 2.95 Å, respectively. The small difference between RMSD values of Heme and CoPP in the antagonist form of the receptor was unable to explain the specific behavior of each ligand.

### 2.5. Dynamic Behavior of REV-ERBβ in the Agonist Form

In the agonist form of the receptor, the stability of the Heme-complex was better than that of the CoPP-complex, with average RMSD values of 4.89 Å and 6.09 Å, respectively. The low RMSD value of the Heme-complex represents the natural dynamics of the crystal structure, indicating the action of the Heme agonist on the receptor (Figure 7b). By contrast, the antagonistic action of CoPP resulted in increases in the RMSD from 40 ns to 200 ns. Furthermore, CoPP, but not Heme, appeared to change the overall conformation of Agonist-REV-ERBβ (Figure 8a,b). Thus, the results suggest that the antagonist ligand might influence the function of the protein target in the agonist form by altering the conformation of the target.

The ligand-binding pocket of REV-ERBβ is composed of His381, Leu382, Val383, Cys384, Phe443, Leu446, Phe450, Phe454, Gly478, Asp481, Leu482, Leu483, and Ser576. The MD trajectory indicated that CoPP moved away from the binding pocket (Figure 8c), whereas Heme did not (Figure 8d). The results also revealed that CoPP rotated about 90° by the end of the simulation.

The movement of CoPP enlarged the cavity of the REV-ERBβ ligand-binding pocket (Figure 9a,b). POVME 3.0 was used to calculate the binding-pocket volume during the MD simulation. Figure 9c shows that the binding-pocket volume of Heme was smaller than that of CoPP, with average values of 566.29 A^3^ and 777.39 A^3^, respectively. 

In the ligand-binding pocket, Cys384 and His568 play important roles in stabilizing Heme through axial coordination with the iron atom. Therefore, a change of the metal atom to cobalt might disrupt that interaction and lead to the expansion of the ligand-binding pocket. The average distance between cobalt and Cys384 (4.72 Å) was shorter than that between iron and Cys384 (5.40 Å). By contrast, the distance between cobalt and His568 was 2.36 Å longer than that between iron and His568 (8.96 Å and 6.60 Å, respectively). The interactions of CoPP and Heme with the two residues are visualized in Figure 10. In the reported crystal structures, the distance was the same from Cys384 to the metal centers of Heme and CoPP (2.34 Å), but the distances from His568 to the respective metal centers differed by 0.14 Å [21]. Nevertheless, the disruption of the ligand-binding pocket still could not explain the antagonistic effect of CoPP on REV-ERBβ. 

The antagonistic effect of the CoPP ligand on REV-ERBβ is based on disruption of the recruitment of transcriptional corepressors. Therefore, the effects of Heme and CoPP on corepressor binding were examined further.

### 2.6. Recruitment of NCoR by Protoporphyrin IX

Protein-peptide docking was conducted to differentiate the NCoR binding to REV-ERBβ in the presence of Heme and CoPP, respectively. The binding affinity of NCoR was evaluated based on atomic contact energy (ACE) and MM/PBSA binding energy. The NCoR structure was taken from a crystal structure with REV-ERBα (PDB ID: 3N00). First, the method validation using the PATCH Dock was performed to determine several parameters that will be used in the protein-peptide docking simulation. The validation stage of this method is carried out using a re-docking approach. In this re-docking process, the values of RMSD and the active site were observed as well as the possible binding residues of the NCoR.

The RMSD value obtained from the re-docking results showed 1.3757 Å (Table 3). Therefore, the parameters of the method validation results can be used at the protein-peptide docking simulation. Table 4 shows that the Heme-complex interacting with NCoR obtained a high PatchDock score with ACE score and binding energy value of −74.60 kJ/mol and −13.33 kJ/mol, respectively. The CoPP-complex interacting with NCoR produced lower binding energy than the Heme-complex, with ACE score and binding energy value of 453.13 kJ/mol and 43.01 kJ/mol, respectively.

The ACE score is an atomic desolvation energy, which is defined as the energy of replacing a protein-atom–water contact with a protein-atom–protein-atom contact [24]. MM/PBSA calculations were performed to evaluate the relative stability of each complex resulting from protein-peptide docking. The results of protein-peptide docking and MM/PBSA showed that the Heme-complex had a better ability to recruit NCoR than the CoPP-complex. That phenomenon appears to be a unique feature of Heme, as synthetic agonists that induce NCoR recruitment in a cellular context cause an increase in corepressor-peptide binding [21].

The interaction between the Heme-complex and NCoR consisted of three hydrogen bonds (with Gly441, Ser499, and Gln529, respectively), one electrostatic interaction (with Thr552), and seven hydrophobic interactions (with Trp402, Val413, Leu482, Leu483, His568, and His578, respectively). The interaction between the CoPP-complex and NCoR consisted of six hydrogen bonds (with Gly441, Ser513, Ser499, Gln529, and His578, respectively) and four hydrophobic interactions (with Val413, Lys439, Leu483, and Arg562, respectively). For that reason, it can be predicted that the positive ACE score and MM/PBSA value for the CoPP-complex were the result of an absence of electrostatic interactions.

The end of Helix-11 is a structural area that is part of the NCoR binding site [25,26]. Compared with the crystal structure, the CoPP-complex undergoes many changes in the conformation of Helix-11 [21]. There were 13 amino acid residues that turned into loops at the end of the simulation, including Ser563, Leu564, Asn565, Asn566, His568, Ser569, Glu570, Glu571, Leu572, Leu573, Ala574, Phe575, and Lys576 (Figure 11b). In contrast to CoPP, that part of Helix-11 in the Heme-complex retained similarities to the crystal structure, with changes at only a few amino acid residues located in the middle of Helix-11 (Pro599, Asp560, Leu561, Arg562, Ser563, Leu564, Asn565, Asn566, and His 568; Figure 11c). Therefore, the Heme-complex was better able to recruit nuclear receptor corepressors and repress the transcription of downstream genes.

Based on the visualization, it could be observed that non-polar peptides derived from NCoR bound to a hydrophobic patch on the LBD of REV-ERBβ. Compared with the crystal structure, there was a difference in the binding area of the corepressor (Figure 12). The changes in conformation at the end of Helix-11 caused by the enlarged CoPP–Agonist-REV-ERBβ-LBD (Figure 13) might weaken and negatively affect corepressor recruitment. Thus, the ligands might control the switch between open and closed conformations of the hydrophobic surface on the LBD for corepressor recruitment [27,28].

## 3. Discussion

It is difficult to describe changes in protein structure that accompany ligand–protein interactions in order to distinguish structural changes that occur when agonists and antagonists interact with the same protein [29]. Even experimental data in the form of crystal structures are unable to distinguish the actions of ligand agonists and antagonists on target proteins [30]. Nuclear receptor REV-ERBβ has two forms of crystal structure, an agonist form (complexed with Heme) and an antagonist form (complexed with CoPP). Heme and CoPP are protoporphyrin IX ligands that have affinity for REV-ERBβ. The only difference between them is in the metal atoms in the middle of the protoporphyrin IX ligand. Heme is a prosthetic group that consists of a heterocyclic protoporphyrin IX ring with an iron ion as the metal center. Heme is an essential component of many proteins, including oxygen transport proteins such as hemoglobin and myoglobin as well as the cytochrome p450 enzymes, in which the Heme moiety carries out electron transport [31,32]. Beyond Heme, an array of other protoporphyrin IX compounds have been synthesized and/or are found naturally in cells, depending on the physiological availability of different metal atoms in tissues. Instead of mimicking the agonist action of Heme, CoPP functions as an antagonist of REV-ERBβ function [21]. That phenomenon was unexpected, because the only distinction between the two ligands is the coordinated metal ion. Therefore, subtle changes in the porphyrin metal center and ring conformation appear to influence the agonist versus antagonist actions of protoporphyrin IX.

In general, protoporphyrin IX ligands have bonds with Val, Gly, and Leu residues in REV-ERBβ. Heme has several additional bonds with Cys and His residues in REV-ERBβ. In the Fe(III) state, a six-coordinate Heme-receptor complex is formed, involving Cys and His residues. Upon reduction to Fe(II), the Cys ligand is replaced by another neutral-donor ligand to form a six-coordinate species, or else the Heme can exist as a five-coordinate species with only the His ligand [21]. That phenomenon allows Heme to be a suitable ligand for REV-ERBβ. In contrast to Heme, CoPP has no interaction with Cys and His residues and therefore has a lower binding energy than Heme. CoPP displays a complex pharmacology, which includes an ability to induce Heme oxygenase 1 (HMOX-1) expression. The induction of HMOX-1 has been suggested as the mechanism by which CoPP displays its anti-obesity activity [33].

There are no changes known to be typical of Antagonist-REV-ERBβ, so the differential characteristics of ligand agonists and antagonists cannot be observed for that species. Heme ligands can stabilize target receptors. Such stabilization is observed in Helix-3 and Helix-11 of REV-ERBβ, which play a role in binding to nuclear receptor corepressors [20,21,34]. REV-ERBs function as transcriptional repressors by recruiting NCoR–HDAC3 complexes to REV-ERB response elements in enhancers and promoters of target genes. REV-ERBβ agonists, such as endogenous Heme or synthetic agonists, function by increasing the transcriptional repression of REV-ERBβ target genes [15,19,21,35]. As with Heme, CoPP does not cause damage to the active site of Antagonist-REV-ERBβ, allowing NCoR binding to REV-ERBβ-LBD. The presence of both ligands is expected to not change the ability of Antagonist-REV-ERBβ to recruit nuclear receptor corepressors and thus repress the transcription of downstream genes [21].

The presence of CoPP in Agonist-REV-ERBβ changes the end part of Helix-11, causing a decrease in the ability of REV-ERBβ to recruit NCoR. In addition, the regions around Helix-11 display the highest B-factors in Agonist-REV-ERBβ. In the CoPP-complex, Helix-11 moves slightly out of the ligand-binding pocket (open conformation), therefore modulating the degree to which the corepressor-binding surface is expanded. Thus, by opening the LBD of REV-ERBβ, CoPP can reduce the ability of NCoR to interact with the active site of REV-ERBβ. NCoR docking simulations revealed that the CoPP-complex has lower energy than the crystal structure of REV-ERBβ. Corepressors require areas with high hydrophobicity. That is reflected in the non-polar nature of the NCoR, which binds the binding pocket of Agonist-REV-ERBβ [21,34,36]. Compared with the CoPP-complex, the Heme-complex has a more compact corepressor-binding surface, which enhances the recruitment of corepressors. The difference in corepressor-binding surface occurs because Helix-3 and Helix-11 in the Heme-complex remain intact as in the crystal structure, with only a slight change in some amino acid residues located in the middle of Helix-11. Thus, the end of the Helix remains intact, and the ability of Heme to recruit nuclear receptor corepressors is not affected. The apparent constitutive repressor effect previously noted for REV-ERBβ is most likely due to the fact that all cells have some level of intracellular Heme [21].

The results of this study can explain the behaviors of agonist and antagonist ligands that bind to REV-ERBβ to recruit nuclear receptor (NR) corepressors. MD simulations showed that Heme and CoPP caused identical changes to Helix-11 of REV-ERBβ, which is one of the key structural elements for ligand binding and corepressor recruitment in REV-ERBβ. It is also evident that subtle changes in the protoporphyrin IX metal center and ring conformation can influence the agonist versus antagonist actions of protoporphyrin IX. That agrees with other studies suggesting that a ligand binding to the iron metal center in Heme drives alterations in REV-ERBβ activity. This study provides new information about the interactions of porphyrins with the cancer suppressor REV-ERBβ and the effects of those interactions on the ability of REV-ERBβ to regulate transcription. Those results will be useful in efforts to develop novel porphyrin-based agents for cancer therapy.

## 4. Materials and Methods

### 4.1. Macromolecule Preparation

Two crystalline structures of nuclear receptor REV-ERBβ were used as target molecules in this study. Both receptor structures were downloaded from the Protein Data Bank with PDB ID 3CQV (Agonist-REV-ERBβ) [20] and 4N73 (Antagonist-REV-ERBβ) [21], respectively. The preparations of macromolecules were conducted by removing the water molecules and natural ligand, adding polar hydrogen atoms, and calculating the Kollman charges.

### 4.2. Ligand Preparation

The ligands used in this study were natural ligands bound to each REV-ERBβ crystal structure. Iron protoporphyrin IX (Heme) was used as an agonist ligand, whereas cobalt protoporphyrin IX (CoPP) was used as an antagonist ligand. Both structures were used as inputs for molecular docking studies (Figure 14).

### 4.3. Molecular Docking Study

The docking study was performed using AutoDock 4.2 with MGLTools 1.5.6 [37,38]. All ligands used in the simulations were set with maximum torsion [39]. The grid spacing was changed from 0.375 nM, and the cubic grid map was 64 × 60 × 60 Å toward the REV-ERBβ binding site. The docking parameters were set as follows: the number of Genetic Algorithm (GA) Runs was set as 100, population size was set as 150, the maximum number of evaluations was set as 2,500,000, and 100 runs were performed. Then, special parameters were prepared for iron (Fe) and cobalt (Co) as the central metal atom of protoporphyrin IX, respectively, and all other parameters were set as the default values. The docking process was performed as follows. First, molecular docking was performed to evaluate the docking poses. Then, defined docking was conducted on the binding pocket. Three to six independent docking calculations were conducted. The corresponding lowest binding energies and predicted inhibition constants (pKi) were obtained from the docking log files (dlg) [40]. AutoDock Tools and Discovery Studio 2016 were used to visualize the docking results. Surface representation images showing the binding pocket of REV-ERBβ were generated using Discovery Studio 2016 software [41].

### 4.4. Molecular Dynamics Simulation

Molecular dynamic simulations were performed for both ligands against each nuclear receptor REV-ERBβ. The simulations were performed using Gromacs2016 [42,43,44,45,46], and the analyses were performed with visual molecular dynamics (VMD, Theoretical and Computational Biophysics group at the Beckman Institute, University of Illinois at Urbana-Champaign) [47]. AMBER99SB-ILDN force field [48,49] was used to parameterize the protein. The ligand was parameterized using two software programs: AnteChamber Python Parser interfacE (ACPYPE) for protoporphyrin IX [50] and ForceGen for the central metal atoms iron (Fe) and cobalt (Co) [51]. Long-range electrostatic force was determined by the Particle Mesh Ewald method [52]. Neutralization of the system was performed by adding Na+ and Cl− ions. The cubic of TIP3P water model was used to solvate the system. Berendsen thermostats and barostats are used during the heating stage with the pressure being maintained at 1 bar. The steps of the simulation included minimization until 500,000 steps, heating until 310 K, temperature equilibration (NVT) in 500 ps, pressure equilibration (NPT) in 500 ps, and a production run with a 2 fs timestep for 200 ns. The stability of the system was verified by analysis of the energy, temperature, pressure, and root-mean-square deviation (RMSD). Analyses of the binding-pocket analysis and cavities were carried out using POVME 3.0 [53].

### 4.5. Protein-Peptide Docking

Protein-peptide docking was carried out between the nuclear receptor corepressor (NCoR) and the structures of the CoPP and Heme complexes obtained from molecular dynamics simulations to observe the ability of each ligand to recruit the corepressor. The NCoR structure that formed in crystals with REV-ERBα was downloaded from Protein Data Bank (PDB ID 3N00) [54] and then separated. PATCH Dock software was used to observe protein–protein interactions based on molecular docking. That software uses the shape complementarity of soft molecular surfaces to generate the best starting candidate solution [55,56]. The default clustering RMSD 2.0 Å was used, and the complex type was chosen to be protein–small ligand. Connolly dot surface representations of the molecules as different components such as convex, concave, and flat patches were generated using the PATCH Dock algorithm [57,58]. Then, the PATCH Dock solutions were optimized for small-size molecules, provides a list of refined complexes, reshuffled the molecule’s relative orientation, and the side-chains interface of the top 10 candidate solutions were rescored. The orientation of the relative molecules were amended by confining the flexibility of the side-chains of the interacting surface and allowing movements of small rigid bodies [59]. The results of the protein-peptide docking simulation were identified based on the PatchDock score, atomic contact energy (ACE) score, and MM/PBSA binding energy. The interactions and binding in the docked conformations in PDB format were visualized using the Discovery Studio 2016 software.

### 4.6. MM/PBSA Calculation

MM/PBSA calculation was performed using the g_mmpbsa package [60] integrated in the Gromacs software. Polar desolvation energy was calculated with the Poisson-Boltzmann equation with a grid size of 0.5 Å. The dielectric constant of the solvent was set to 80 to represent water as the solvent [61,62]. Non-polar contribution was determined by calculation of the solvent-accessible surface area with the radii of the solvent as 1.4 Å [63]. The binding free energy of the complex was determined based on 200 snapshots taken from the beginning to the end of the molecular dynamic simulation trajectories of the complex.

## 5. Conclusions

The mechanism of action of the agonist of iron protoporphyrin IX (Heme) and the antagonist of cobalt protoporphyrin IX (CoPP) against the REV-ERBβ nuclear receptor can be predicted based on the results of molecular dynamics simulations. The replacement of the central metal atom has been shown to affect the affinity of the two porphyrins to the REV-ERBβ receptor and changes to the protein conformation during molecular dynamics simulations. This phenomenon will affect the ability of the REV-ERBβ receptor to bind to the NCoR corepressor which plays a role in regulating cancer cell signals to suppress target gene transcription. Therefore, in silico studies are very important in designing and developing porphyrin derivatives, especially for the therapy of cancer.

## Figures and Tables

**Figure 1 molecules-26-03251-f001:**
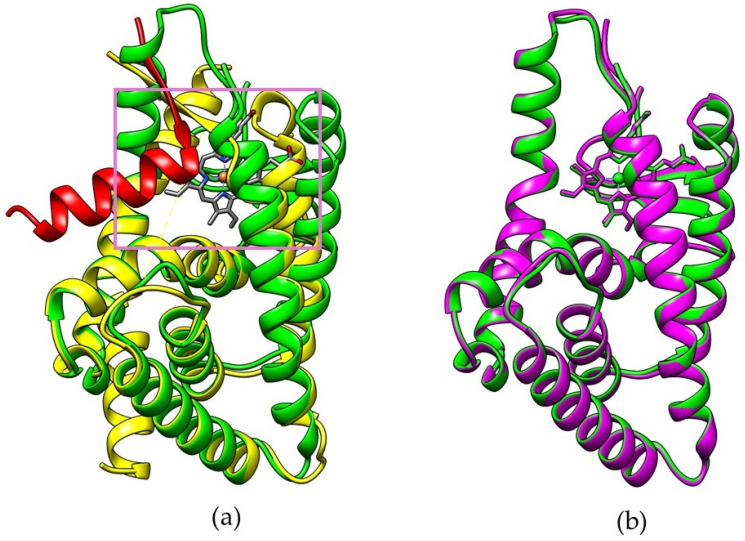
(**a**) Overlay visualization of apo-structure REV-ERBβ (yellow) with bound NCoR (red) and REV-ERBβ in complex form with Heme (green). (**b**) Visualization of REV-ERBβ in complex form with Heme (green) and with CoPP (purple).

**Figure 2 molecules-26-03251-f002:**
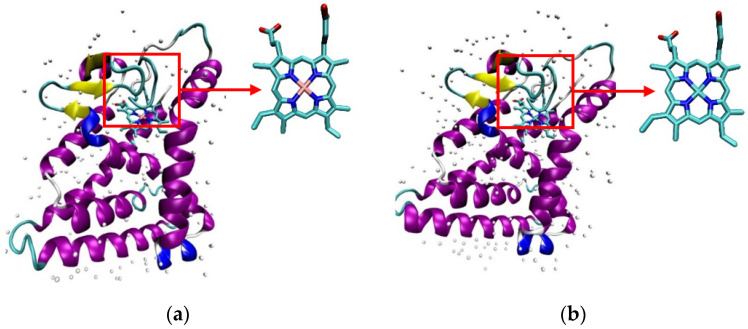
(**a**) The crystal structure of the REV-ERBβ that forms a complex with iron protoporphyrin IX (Agonist-REV-ERBβ). (**b**) The crystal structure of REV-ERBβ that forms a complex with cobalt protoporphyrin IX (Antagonist-REV-ERBβ).

**Figure 3 molecules-26-03251-f003:**
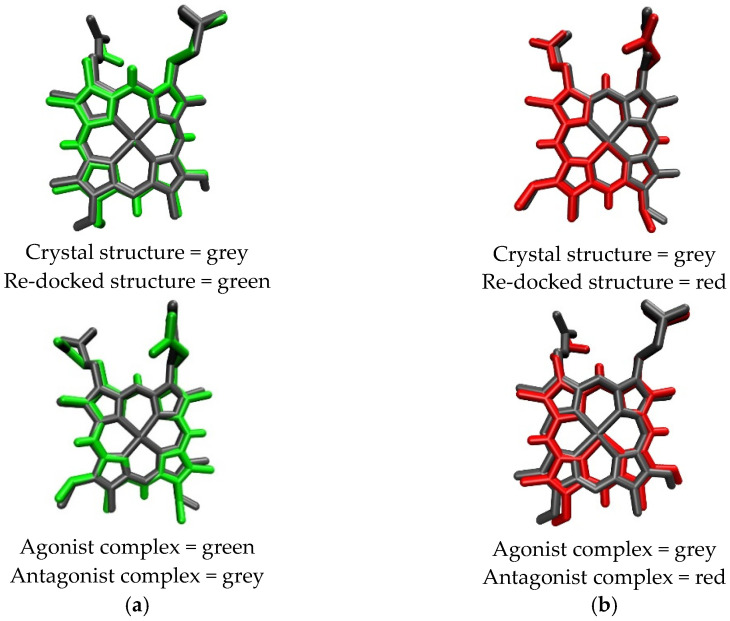
Overlay structures of Heme (**a**) and CoPP (**b**) in re-docking and docking simulation.

**Figure 4 molecules-26-03251-f004:**
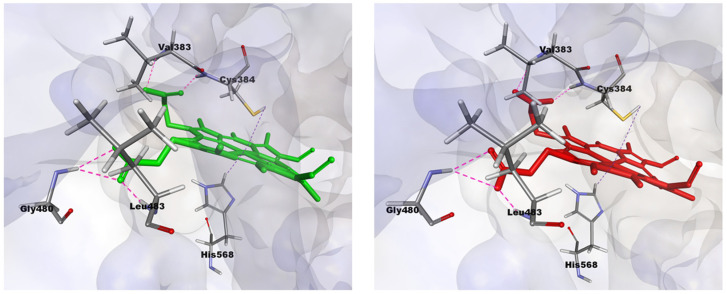
Visualization of interactions occurring in the binding pocket of REV-ERBβ when docked against Heme (green) and CoPP (red).

**Figure 5 molecules-26-03251-f005:**
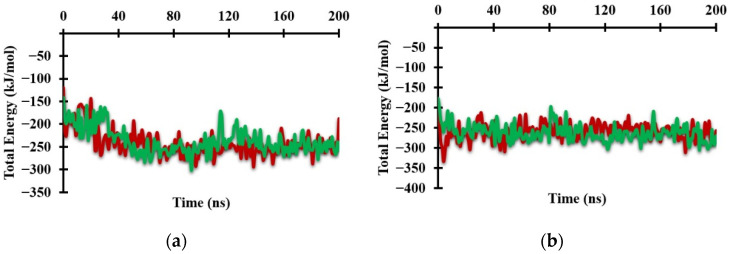
Graph of the total system energy from MM/PBSA calculations when (**a**) Antagonist-REV-ERBβ and (**b**) Agonist-REV-ERBβ form complexes with Heme (green) and CoPP (red) during molecular dynamic simulations.

**Figure 6 molecules-26-03251-f006:**
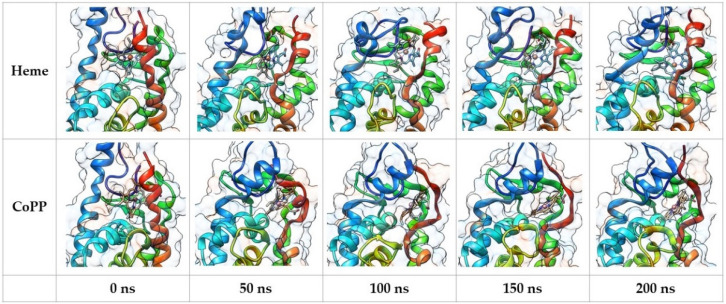
Snapshot of the molecular dynamic simulation trajectories of CoPP and Heme against Agonist-REV-ERBβ.

**Figure 7 molecules-26-03251-f007:**
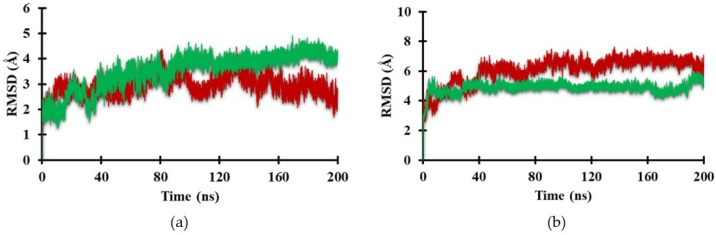
Plot of the RMSD value of backbone in the active site of (**a**) Antagonist-REV-ERBβ and (**b**) Agonist-REV-ERBβ with Heme (green) and CoPP (red) during the molecular dynamic simulation.

**Figure 8 molecules-26-03251-f008:**
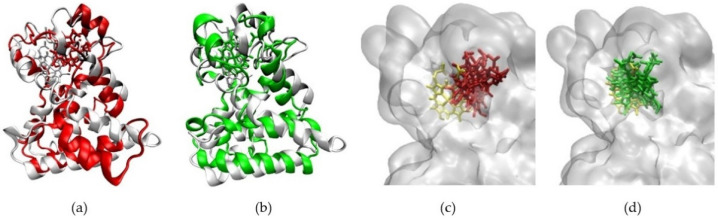
(**a**) The changes in conformation of Agonist-REV-ERBβ (red) when forming a complex with CoPP compared with the initial simulation pose (white). (**b**) The other conformation of Agonist-REV-ERBβ (green) when forming a complex with Heme compared with the initial simulation pose (white). (**c**) Graph of RMSD values of CoPP (red) and Heme (green) in the binding pocket of Agonist-REV-ERBβduring molecular dynamics simulations. (**d**) CoPP and Heme conformations at the end of the simulations.

**Figure 9 molecules-26-03251-f009:**
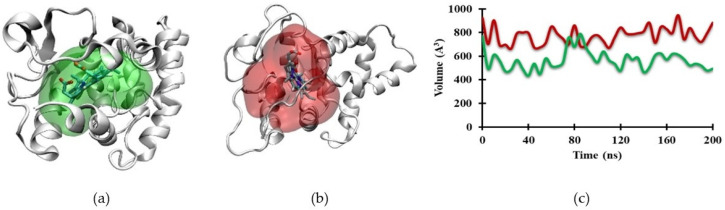
The cavity of (**a**) Heme and (**b**) CoPP on the Agonist-REV-ERBβ binding pocket at the end of the simulation. (**c**) Plots of cavity values of Heme (green) and CoPP (red) in the binding site of Agonist-REV-ERBβ during molecular dynamics simulations.

**Figure 10 molecules-26-03251-f010:**
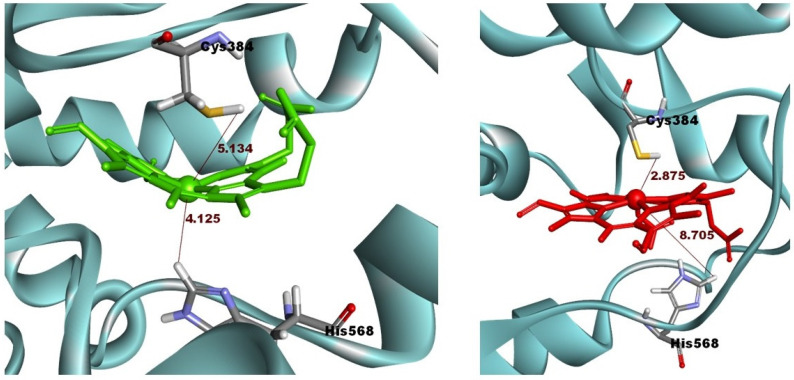
The distance between Cysteine 384 and Histidine 568 on the Agonist-REV-ERBβ and the metal central atom of Heme (green) and CoPP (red) during molecular dynamic simulations.

**Figure 11 molecules-26-03251-f011:**
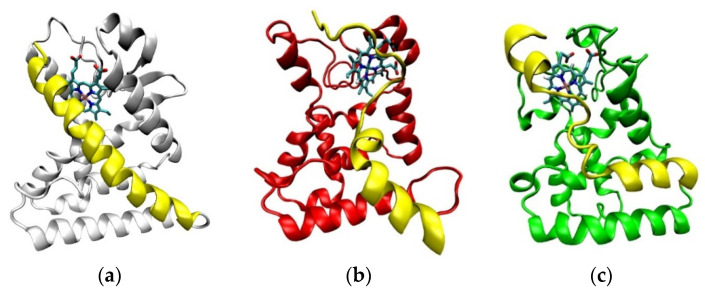
The visualization of Helix-11 (yellow) in Agonist-REV-ERBβ on the crystal structure (**a**), after molecular dynamic simulation in complex with CoPP (**b**) and in complex with Heme (**c**).

**Figure 12 molecules-26-03251-f012:**
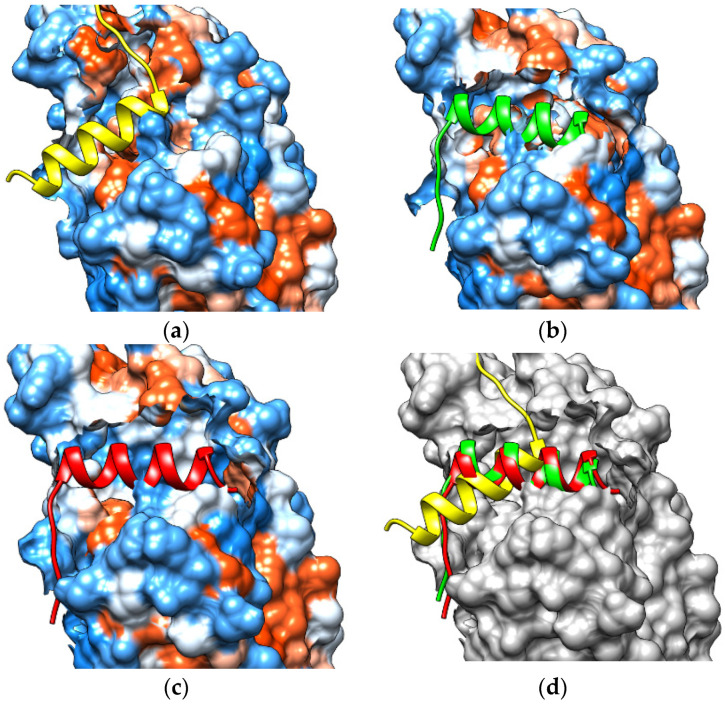
Visualization of the peptide-docking results between the nuclear receptor corepressor (NCoR) and (**a**) the crystal structure, (**b**) the CoPP-complex, (**c**) the Heme-complex, and (**d**) overlay NCoR in crystal structure (yellow), CoPP-complex (green), and Heme-complex (red).

**Figure 13 molecules-26-03251-f013:**
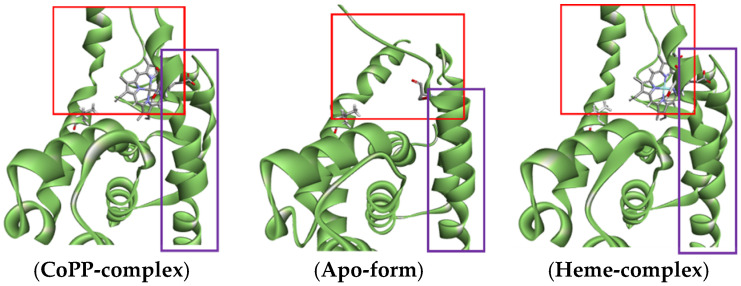
Visualization of the changes in conformation at the end of Helix 11 (purple box) that caused by the enlarged ligand-Agonist-REV-ERBβ-LBD (red box). The visualization of REV-ERB apo-structured was shown as comparison and the NCoR structure was omitted to make clear image.

**Figure 14 molecules-26-03251-f014:**
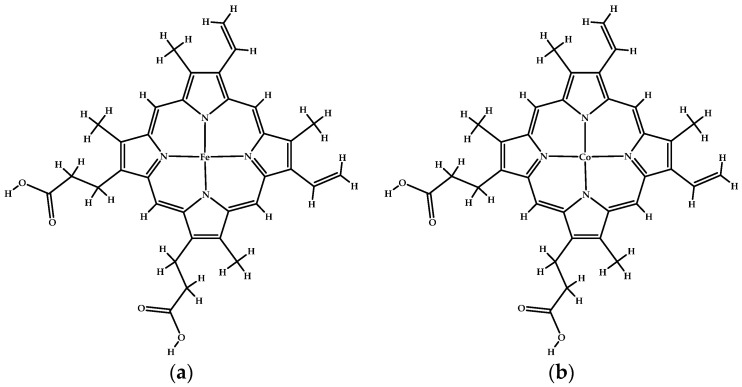
(**a**) The structure of iron protoporphyrin IX (Heme). (**b**) The structure of cobalt protoporphyrin IX (CoPP).

**Table 1 molecules-26-03251-t001:** The binding energy and RMSD value of CoPP and Heme in molecular docking.

Protoporphyrin Ligand	Re-Docking	Docking
Heme	RMSD = 0.3078 Å ∆G (agonist) = −71.10 kJ/mol	RMSD = 0.3703 Å ∆G (antagonist) = −65.91 kJ/mol
CoPP	RMSD = 0.3540 Å ∆G (antagonist) = −56.87 kJ/mol	RMSD = 0.7110 Å ∆G (agonist) = −56.95 kJ/mol

**Table 2 molecules-26-03251-t002:** The free binding energies and their corresponding components for protoporphyrin IX binding to REV-ERBβ.

Complex REV-ERBβ	∆*E*_vdw_ (kJ/mol)	∆*E*_ele_ (kJ/mol)	∆*G*_PB_ (kJ/mol)	∆*G*_NP_ (kJ/mol)	∆*G*_Bind_ (kJ/mol)
Heme-Agonist-REV-ERBβ	−309.61	−208.23	284.17	−28.72	−262.38
CoPP–Agonist-REV-ERBβ	−283.36	−203.37	255.76	−27.33	−258.30
Heme–Antagonist-REV-ERBβ	−300.41	−170.95	261.51	−27.24	−237.10
CoPP–Antagonist-REV-ERBβ	−307.76	−162.81	257.03	−27.26	−240.79

Note: ∆*E*_vdw_ = van der Waals contribution, ∆*E*_ele_ = electrostatic contribution, ∆*G*_PB_ = polar contribution of desolvation, ∆*G*_NP_ = non-polar contribution of desolvation.

**Table 3 molecules-26-03251-t003:** Results of the validation of the protein-peptide docking method.

Amino Acid Residue	RMSD
Trp402, Val413, Lys439, Gly441, Leu482, Leu483, Ser499, Ser513, Gln529, Thr552, Arg562, His568, His578	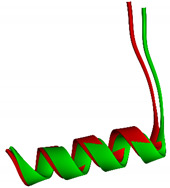 1.3757 Å Crystal structure = red Re-docked structure = green

**Table 4 molecules-26-03251-t004:** Results of protein-peptide docking against nuclear receptor corepressor (NCoR).

Complex REV-ERBβ	Result		
	Score	ACE (kJ/mol)	∆*G*_Bind_ (kJ/mol)
Crystal Structure Agonist-REV-ERBβ	11332	−1903.89	−400.43
Heme−Agonist-REV-ERBβ	9882	−74.60	−13.33
CoPP−Agonist-REV-ERBβ	10424	453.13	43.01

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
