# Peer review of "Molecular Dynamics of Cobalt Protoporphyrin Antagonism of the Cancer Suppressor REV-ERBβ"

_molecules, 2021, doi:10.3390/molecules26113251_

Round 1
Reviewer 1 Report
In the manuscript entitled “Molecular Dynamics of Cobalt Protoporphyrin Antagonism of the Cancer Suppressor REV-ERBβ” the authors attempt to reveal in silico the potential mechanism of action of the agonist, Heme (iron protoporphyrin IX), and antagonist, CoPP (cobalt protoporphyrin IX) to the REV-ERBβ receptor. The authors combined molecular docking and molecular dynamics simulations on the studied systems in order to reveal the reverse effect of the cobalt atom when iron is replaced in Heme on the REV-ERBβ receptor. The study is interesting and actual. Some major revisions are needed:
A re-docking of nuclear receptor corepressor (NcoR) to REV-ERBα in order to validate the docking procedure needs to be performed as the corresponding RMSD value between the re-docked pose and the crystal structure of the peptide should be provided.
Detailed information of the optimization, refinement and reshuffling using the cited PIPER-FlexPepDock server of the derived docking poses for the NcoR peptide from the PATCH Dock server outputs should be included. How were the final poses selected?
The shown poses resulted from the peptide-protein docking on figure 10 b) and c) look quite similar. A discussion of this should be added.
The derived results from MD simulations indicate that only in the case of Agonist-REV-ERBβ system CoPP altered the binding site while this is not observed in the case of the antagonist form of REV-ERBβ. A discussion of the possible reasons for these results should be added.
It is recommended that the molecular dynamic (MD) simulations are performed in saline.
A detailed explanation of molecular dynamics protocol should be provided. It is recommended that an equillibration step before the production dynamics be added. The thermostat and barostat used should be described. The time for each pre-production step needs to be indicated.
What are the special parameters for iron and cobalt during the docking calculations of the ligands to the REV-ERBβ?
What are the Heme and CoPP RMSD values between the crystal structures of Agonist-REV-ERBβ and Antagonist-REV-ERBβ and the corresponding re-docked poses?
In the annotation of Figure 4 it should be denoted which two of the four studied systems are presented.
Reviewer 2 Report
The article submitted by Daryono Hadi Tjahjono and co-workers described the study of molecular dynamics of a cobalt-based porphyrin and, especially, their interaction with receptor REV-ERBbeta.
The authors have provided a complete study on the simulated effects of both heme and Co-TPP bindings onto the nuclear receptor REV-ERB beta. This receptor is involved in several physiological and pathological processes.
Regarding to the potential interests of these structures for the future development of anticancer agents, this paper deserves to be published in Molecules, after some major modifications.
The introduction on heme must be described more deeply, as this chemical structure is not only the “pigment in red blood cells” but is also implied in numerous functions in the living organisms: dioxygen carrier but also xenobiotics metabolism, with the CYP450 superfamily.
After an introduction on the interest of protoporphyrins for the imaging and treatment of cancers, the authors have focused their strategy on the binding of such structures on REV-ERB nuclear receptors.
REV-ERBbeta receptor is overexpressed in some cancer cell lines and appeared so to be of real interest as the binding of heme on this receptor is reported to decrease cancer cells proliferation.
As the readers could not be specialists of that kind of receptors, I strongly suggest the authors to add an additional figure in their paper (in the introducing part). This is an important remark of the reviewer as this will shed some light on the therapeutic interest of their strategy.
It is also known that Heme is a natural agonist of this receptor, whereas some other synthetic products are described in the literature to be agonists.
The objective of this paper is to provide some rationale on the antagonist role of cobalt ProtoPorphyrin CoPP, by studying the molecular dynamics of this complex with REV-ERB beta protein, compared to natural Heme.
To do this, authors followed a well-defined approach, by: 1) understanding the binding modes/energies of Heme versus CoPP within antagonist/agonist REV-ERB beta receptors, 2) determine their free binding/total energies from MM/PBSA method, 3) proceed a structural analysis of both complexes from molecular dynamics simulation, 4) made a special focus on the dynamic behaviour of REV-ERB beta in the antagonist and in the agonist forms.
The molecular modelling study was also carried out more specifically on Metal-PP binding site/pocket (by measuring the distances between the metal complex and key amino acids Cys384 and His568, by comparison with literature data [ref 17]), but, despite marked differences between the two metallic complexes, could not explain the agonist/antagonist effects.
Consequently, in a last part, the authors have studied the recruitment of transcriptional corepressor NCoR by those protoporphyrins IX.
Especially, positions of helix-11 in MePP-REV-ERB beta complexes were considered, followed by the docking of the peptide corepressor in the whole complexes.
Please indicate clearly in the title of Figure 9 the studied protein and what is based on X-ray structures/molecular modelling.
In the same idea, indicate in the title of Figure 10 the nature of the docked peptide (NCoR corepressor?) as this will avoid misunderstandings. A zoom should be done of the peptide binding area to estimate the differences between figures 10b and 10c.
Please indicate all the reference information, when citing some crystallographic files (eg “PDB ID: 3N00” must cite the corresponding literature references).
These studies are appropriately conducted and the conclusions given in the discussion part seems to be relevant, thus justifying the publication of this work.
Please indicate the .pdb file numbers in the subtitles of ALL your figures.
When used for the first time, abbreviation must be defined: for example, I guess that LBD means Ligand Binding Domain but this must be clearly stated.
The structures of iron and cobalt protoporphyrin IX (Heme and CoPP) in Figure 11 are not correct. Some hydrogen atoms are missing, and oxygen atoms in the acid functionalities are missing.
Please write correctly reference 3 as it seems to be a thesis manuscript…
Reference 28 is not written correctly for a book: please modify.
Reviewer 3 Report
In this manuscript, the authors report the molecular dynamics of Heme and CoPP binding to REV-ERBβ to develop new porphyrin-based agents for early cancer detection. Based on the REV-ERBβ deposited in the PDB, the author analyzed the structure through various simulations, and I believe that some of the research results are valuable for future research. However, no biochemical studies have been conducted to prove the results of the author's computer analysis. As a result, it is difficult to support the author's scientific claims for the analysis. Accordingly, the current version of manuscript is not recommended for publication. Instead, I suggest reviewing the paper after some biochemical studies or concerns have been resolved.
1.L102-105: “Iron has an atomic radius of 0.126 nm, whereas cobalt has atomic radius of 0.200 nm, indicating that the radius of the atom at the metal center plays a role in determining the affinity of the porphyrin complex for REV- ERBβ.” It is a difficult argument to understand that the atomic radius determines affinity. What is the basis for the claim?
2. Line 110-112: “In general, CoPP and Heme had the same interactions with the amino acid residues of REV-ERBβ (i.e. Val383, Gly480, and Leu483); however, Heme had additional interactions with Cys384 and His568. ”Is the description based solely on the docking result? I think this claim can be verified through mutagenesis or spectroscopy experiments. It is difficult to trust without experimental evidence.
3. Figure 1, 4: It is helpful to understand the overall structure, but since Heme and CoPP are the focus in the content of this study, it will be helpful to the reader to show the structure at close distance.
4. Line 262-263: “Compared with the crystal structure, the CoPP complex undergoes many changes in the conformation of Helix-11.” I think this large structural change can be verified through circular dichroism. It is difficult to trust without experimental evidence.
5. Line 177-179: “Based on the visualization, it could be observed that non-polar peptides derived from NCoR bound to a hydrophobic patch on the LBD of REV-ERBβ. Compared with the crystal structure, there was a difference in the binding area of the corepressor (Figure 10).” I think this claim can be verified through mutagenesis experiments. It is difficult to trust without experimental evidence.
Round 2
Reviewer 1 Report
I accept the author’s responses but two revisions remain unclear/misunderstood:
How were the PIPER-FlexPepDock docking solutions optimized, refined and reshuffled before being rescored?
What are the special parameters for iron and cobalt during the docking calculations of the ligands to the REV-ERBβ?
Reviewer 2 Report
I have considered the revised version of the work submitted by D.H. Tjahjono & co-workers and I have noticed some efforts of the authors to answer my recommendations.
However, despite these efforts, I have also noticed that some of my comments were not considered by the authors, though I have indicated there were “important”: “This is an important remark of the reviewer as this will shed some light on the therapeutic interest of their strategy.”
1) The introduction on heme must be described more deeply, as this chemical structure is not only the “pigment in red blood cells” but is also implied in numerous functions in the living organisms: dioxygen carrier but also xenobiotics metabolism, with the CYP450 superfamily.
2) As the readers could not be specialists of that kind of receptors, I strongly suggest the authors to add an additional figure in their paper (in the introducing part). This is an important remark of the reviewer as this will shed some light on the therapeutic interest of their strategy.
3) When used for the first time, abbreviation must be defined: for example, I guess that LBD means Ligand Binding Domain but this must be clearly stated.
So, please correct your manuscript accordingly.
Reviewer 3 Report
Thank you for the author's sincere response to the reviewer's concerns in the revised manuscript. I think this manuscript is definitely academically valuable, but there are still concerns about where no experimental evidence has been provided. In my personal opinion, experimental results may be reliable, but computational studies may or may not. I am not sure if my standards are strict in terms of computational study. Accordingly, I would like to follow the judgment of other reviewers or editors regarding the recommendation of revised manuscript for the publication.
Meanwhile, the revised manuscript needs some improvements.
- Table 1: It seems to be appropriate to move the main or supplementary figure rather than the figure in the table.
- Figure 2: It is necessary to increase the size of the label for amino acids in the figure. In addition, I think that the ratio for Heme and CoPP molecules, or orientation of the protein needs to be the same.
- Figure 3, 5 and 7: The axis indicating the value of the figure and the number are displayed in black, and the text size in figure should be increased.
- Figure 4: Cartoon styles for Heme and CoPP-bound structure should be same.
- Figure 8: Labels for amino acids and distance markings should be large.
